# Continuity and sustainability of care in family medicine: Assessing its association with quality of life and health outcomes in older populations—A systematic review

**Mohammed Nasser Albarqi** *

Associate Professor of Family Medicine, College of Medicine, King Faisal University, Hofuf, Saudi Arabia

* aalbarqi@kfu.edu.sa

## Abstract

### Background

Continuity of care is a core principle of family medicine associated with improved outcomes. However, fragmentation challenges sustaining continuous relationships. This review aimed to provide timely and critical insights into the benefits of continuity and sustainability of care for older adults.

### Methods

PubMed, EMBASE, CINAHL, Cochrane Library were systematically searched for studies on continuity/sustainability models in family medicine and effects on older adults. 14 studies met inclusion criteria for final synthesis. Quality was assessed using ROBINS-I. Outcomes were narratively and thematically synthesized.

### Results

Greater continuity of care was consistently associated with reduced healthcare utilization including lower emergency department visits and hospitalizations. Continuity also correlated with improved chronic disease management, care coordination, patient-reported experiences, and quality of life. Patient-centered medical homes and care coordination models showed potential to strengthen continuity and sustainability. Thoughtful telehealth integration and technology tools augmented continuity.

### Conclusion

Continuous healing relationships are vital for patient-centered care of older adults. While current fragmentation challenges sustainability, innovations in primary care teaming, coordination, telehealth, and health information technology can extend continuity's benefits. Realizing improvements requires system-wide reorientation toward relationships and whole-person care.

**Data Availability Statement:** All relevant data are within the manuscript and its Supporting Information files.

**Funding:** This research was funded by the Deanship of Scientific Research at King Faisal University, Saudi Arabia (GRANT6090). The funders had no role in study design, data collection and analysis, decision to publish, or preparation of the manuscript.

**Competing interests:** The authors declare that there are no conflicts of interest regarding the publication of this paper.

## 1. Introduction

In the realm of family medicine, sustainability is a critical concept that encompasses the capacity of healthcare systems to provide consistent, high-quality care over time, particularly for the aging population [1]. The sustainability of continuity of care in family medicine implies the ability to offer uninterrupted, coordinated, and comprehensive care across different levels and sites of care, adapting to the evolving needs of older populations [2]. It involves ensuring that healthcare practices are both effective in promoting health outcomes and efficient in utilizing resources to maintain these standards over the long term [3]. Sustainable care models focus on preventative care, management of chronic conditions, and integration of services, aiming to reduce hospital readmissions, minimize healthcare costs, and improve the overall quality of life for older individuals [4]. By emphasizing a sustainable approach, family medicine can better address the complex, multifaceted health needs of the elderly, promoting longer, healthier lives and resilience in healthcare systems [5].

The integration of sustainability into the continuity of care reflects a commitment to not only immediate health outcomes but also the long-term well-being of older populations [6]. Sustainable practices in family medicine involve training and retaining skilled healthcare professionals, leveraging technology for efficient service delivery, and fostering patient-centered care that respects the preferences and values of older adults [7]. It also means advocating for policies and research that support long-term investments in community health infrastructure, innovative care models, and health education [8]. These efforts ensure that quality care is not just available today but is also adaptable and resilient to meet the future health challenges of aging populations [9]. Ultimately, incorporating sustainability into continuity of care is about creating a legacy of health and well-being for older generations, ensuring they have access to the care and support they need to thrive in their later years [10].

The concept of continuity of care has long been a core tenet of family medicine and primary care [11]. Continuity involves an ongoing caring relationship between a patient and clinician over time, rather than isolated episodic encounters [12]. This longitudinal partnership facilitates the provider gaining comprehensive knowledge of the patient's medical history, values, preferences, and goals. Continuity of care has been linked to a multitude of benefits including improved delivery of preventive services, enhanced chronic disease management, increased patient trust in providers, improved patient and provider satisfaction, and reductions in healthcare utilization and costs [13–15].

For older patients with multiple chronic conditions, continuity with a family physician may bring particular benefits for care coordination, patient-centered care, and overall well-being [16, 17]. However, continuity has declined substantially over recent decades across many healthcare systems, due to fragmentation in care delivery, increased patient mobility, growth of specialty services, electronic medical records implementation, and insurance changes [18–20]. Reestablishing and improving continuity of care for older adults represents a priority for professional organizations like the American Academy of Family Physicians and policymakers in order to enhance outcomes for the aging population [21].

Continuity of care has been described as consisting of three main dimensions: informational, management, and relational continuity [22]. Informational continuity refers to the provider having access to the patient's medical history and prior care events through medical records and data sharing across settings [23, 24]. Management continuity means consistent evidence-based care in alignment with patient goals and values [25]. Relational continuity involves the longitudinal personal relationship between the patient and one or more clinicians built on accumulated trust and familiarity [26]. While interrelated, these dimensions can be

disrupted independently, necessitating a comprehensive approach to evaluating and strengthening continuity of care [27, 28].

Given its multidimensional nature, measuring continuity of care poses challenges. Approaches used in research have included survey questions, scoring systems leveraging usage data, and direct observations of patient-provider encounters [29, 30]. Asking patients if they have a regular doctor whom they see for most of their care has been a commonly used and validated survey item to simply assess informational continuity [31]. More complex validated measures like the Usual Provider of Care index quantify continuity over time using visit patterns, though require extensive clinical data [32].

Several systematic reviews have aggregated the broad evidence linking greater continuity of primary care to lower mortality, reduced hospitalizations, lower costs, and higher patient satisfaction [33, 34]. However, the benefits of continuity may be greatest for older adults given their higher needs for care coordination and chronic disease management [35]. For seniors specifically, greater continuity has been associated with higher receipt of recommended preventive cancer screening and immunizations [36]. Among older adults with chronic conditions like diabetes, continuity relates to improved disease control through regular primary care monitoring [37]. For low-income seniors, having a regular physician correlates to lower likelihood of unmet needs and emergency department use [38, 39].

In regard to hospital care, a landmark study found elderly patients seeing the same primary care physician had lower rates of hospitalizations for ambulatory care-sensitive (ACS) chronic conditions like COPD, suggesting timely outpatient care avoided costly acute care [40, 41]. Similarly, another analysis found greater primary care continuity linked to reduced 30-day readmissions among chronically ill seniors [33, 42]. Having a continuous partnership with a provider who understands patients' goals may facilitate transitions home and prevent complications requiring rehospitalization [43].

This evidence indicates continuity in the outpatient setting may enable proactive disease management for older patients with complex needs, avoiding downstream emergency and inpatient services [44]. However, gaps remain regarding continuity's impact on critical patient-centered outcomes like functional status, quality of life, and satisfaction with care. Assessing continuity's role in overall well-being is increasingly essential given the focus on holistic care for aging populations [45, 46].

Initiatives such as the patient-centered medical home model have sought to promote sustained, high-quality relationships between older patients and primary care providers [47]. Core features of medical homes include enhancing access through expanded hours, care coordination using integrated teams, and patient engagement in shared decision-making [48]. By offering these augmented services, the medical home aims to facilitate longitudinal relationships over time.

Evidence suggests medical homes may improve continuity for seniors. One study found patients at small practices transformed into medical homes reported growth in continuity compared to the control group [49]. Another analysis found joining a medical home correlated to increased annual visits, suggesting improved continuity, for patients with diabetes [37]. Overall, the integration of services and patient-centered focus inherent in the medical home shows potential to strengthen continuity of care for older individuals.

This systematic review aimed to provide timely and critical insights into the benefits of continuity and sustainability of care for older adults. Findings may support initiatives to improve continuity within age-friendly medical homes and help shape the implementation of emerging care models. Results can inform training of current and future family physicians on best practices in cultivating longitudinal relationships with older patients. This review aimed to identify remaining knowledge gaps to guide health services researchers. Ultimately, optimizing

continuity of care will lead to a healthier aging population, improved quality of life for seniors, and more responsible use of finite healthcare resources.

## 2. Materials and methods

### 2.1 Search strategy and selection criteria

This systematic review adheres to the Preferred Reporting Items for Systematic Reviews and Meta-Analyses (PRISMA) guidelines to ensure a rigorous, transparent, and comprehensive methodology [50]. The research protocol, in line with PRISMA-P (Preferred Reporting Items for Systematic Reviews and Meta-Analysis Protocols) statement recommendations [51] and registered with PROSPERO to provide an accessible and detailed account of the review process (CRD42023491239).

A comprehensive search strategy was implemented to explore the literature related to continuity and sustainability of care in family medicine and its impact on the quality of life and health outcomes of older populations. Key databases such as PubMed, MEDLINE, Embase, Web of Science, Cochrane Library, and Scopus were systematically searched. The strategy incorporated a mix of Medical Subject Headings (MeSH) and free-text keywords tailored to capture the broadest spectrum of relevant studies. Terms included "Family Medicine," "Continuity of Care," "Sustainable Healthcare," "Quality of Life," "Health Outcomes," and "Older Populations.". the detailed search strategy described in the Table 1.

The search aimed to be as inclusive as possible while retaining specificity to the topic of interest. It was finalized on a specific date to ensure a current and comprehensive collection of relevant literature.

**Table 1. The detailed search strategy.**

| Database | Search Terms |
|---|---|
| PubMed | ("Continuity of Patient Care"[Mesh] OR "continuity of care" OR "care continuity") AND ("Family Practice"[Mesh] OR "family medicine") AND ("Quality of Life"[Mesh] OR "life quality" OR "health-related quality of life") AND ("Aged"[Mesh] OR "older adults" OR "elderly" OR "senior") AND ("Health Outcomes"[Mesh] OR "health results" OR "patient outcome*" OR "treatment outcome*") AND ("Sustainability"[Mesh] OR "sustainable practices" OR "long-term care") |
| MEDLINE | Same as PubMed |
| Embase | 'continuity of patient care'/exp OR 'continuity of care' OR 'care continuity' AND 'family practice'/exp OR 'family medicine' AND 'quality of life'/exp OR 'life quality' OR 'health-related quality of life' AND 'aged'/exp OR 'older adults' OR 'elderly' OR 'senior' AND 'health outcomes'/exp OR 'health results' OR 'patient outcome*' OR 'treatment outcome*' AND 'sustainability'/exp OR 'sustainable practices' OR 'long-term care' |
| Web of Science | TS = (continuity of patient care OR continuity of care OR care continuity) AND TS = (family practice OR family medicine) AND TS = (quality of life OR life quality OR health-related quality of life) AND TS = (aged OR older adults OR elderly OR senior) AND TS = (health outcomes OR health results OR patient outcome* OR treatment outcome*) AND TS = (sustainability OR sustainable practices OR long-term care) |
| Cochrane Library | 'continuity of care' in Title Abstract Keyword OR 'family medicine' in Title Abstract Keyword AND 'quality of life' in Title Abstract Keyword OR 'older populations' in Title Abstract Keyword AND 'health outcomes' in Title Abstract Keyword AND 'sustainability' in Title Abstract Keyword |
| IEEE Xplore | ("continuity of care" OR "care continuity") AND "family medicine" AND ("quality of life" OR "life quality") AND ("older adults" OR "elderly" OR "senior") AND ("health outcomes" OR "patient outcome*" OR "treatment outcome*") AND "sustainability" |
| Scopus | TITLE-ABS-KEY (continuity of care OR care continuity) AND TITLE-ABS-KEY (family medicine) AND TITLE-ABS-KEY (quality of life OR life quality OR health-related quality of life) AND TITLE-ABS-KEY (aged OR older adults OR elderly OR senior) AND TITLE-ABS-KEY (health outcomes OR patient outcome* OR treatment outcome*) AND TITLE-ABS-KEY (sustainability OR sustainable practices OR long-term care) |

Following the initial search, duplicates were removed, and the remaining articles were screened based on title, abstract, and full-text assessment. Inclusion criteria focused on studies specifically investigating continuity and sustainability in family medicine and its correlation with quality of life and health outcomes in older adults. Exclusion criteria included studies not pertinent to the research questions, non-English language studies, reviews, conference abstracts, and those lacking original data. Any discrepancies during the selection process were resolved through discussion and consensus among the review team.

## 2.2 Data extraction

Data extraction aimed to gather comprehensive information pertinent to the continuity and sustainability of care in family medicine. This included study characteristics, descriptions of care models, patient demographics, quality of life metrics, and health outcomes. The review prioritized extracting data that directly correlated with the older population's experience with continuous and sustainable care models. Authors of the primary studies were contacted as necessary for additional data or clarification. In the initial search through the databases, a total of 5316 papers were found. After removing duplicates, 485 papers were screened based on their title and abstract, with 213 being excluded. Of the remaining 185papers, 14 were ultimately selected for the full-text review. The PRISMA flow diagram is provided in (S1 Fig).

## 2.3 Quality assessment

The methodological quality and risk of bias of all included studies were rigorously evaluated using a modified version of the ROBVIS2 tool. ROBVIS2 it was developed during the *Evidence Synthesis Hackathon*, This web app is built on the ROBVIS R package [52]. Discrepancies in quality assessments were resolved through discussion to reach a consensus among the review team.

**2.3.1 Assessment of potential biases in reviewed articles.**   In our systematic review, we meticulously assessed potential biases in the included articles using the ROBINS-I tool for non-randomized interventions and the Cochrane Collaboration's tool for randomized trials, where applicable. This comprehensive evaluation covered selection bias by examining participant selection methods and baseline comparability, performance bias through the implementation of interventions and blinding of participants and providers, detection bias by assessing the blinding of outcome assessors and uniform application of outcome measures, and attrition bias by analyzing follow-up completeness and data handling. Reporting bias was also scrutinized to ensure all expected outcomes were duly reported, alongside a consideration of publication bias to counteract the tendency for publishing studies with positive results preferentially. This thorough assessment of biases was crucial for ensuring the reliability and validity of our findings, providing a clear, critical evaluation of the methodological quality of the included studies, and highlighting areas necessitating further investigation to enrich our understanding of the impact of continuity and sustainability of care on the health outcomes and quality of life in older populations.

## 2.4 Data analysis

The review employed both narrative and thematic analysis methods to synthesize the data:

**Narrative Synthesis:** This qualitative approach provided a detailed summary and interpretation of the findings from the included studies. It emphasized understanding the various aspects of continuity and sustainability in family medicine and their effects on the quality of life and health outcomes in older populations. The narrative synthesis aimed to highlight significant trends, differences, and implications for healthcare providers and policymakers.

**Thematic Analysis:** Through a systematic identification of themes and patterns across the studies, this method facilitated a deeper understanding of the nuances and variations in care models. It involved coding and categorizing findings to explore the effectiveness, challenges, and best practices related to sustaining continuous care in family medicine. Thematic analysis allowed for a more nuanced and detailed exploration of the data, providing insights into the complexities and dynamics of sustainable healthcare for older adults.

Through these methodologies, the review aims to provide a comprehensive and nuanced understanding of how continuity and sustainability in family medicine affect the quality of life and health outcomes of older populations, offering valuable insights and recommendations for practice and policy.

## 3. Results

### 3.1. The quality assessment

The risk of bias assessment for the included studies [53–66] presents a varied landscape regarding the quality and reliability of the evidence provided (S2 Fig). The study by Asma et al., 2020, stands out with a low risk across all domains, suggesting a high level of methodological rigor and reliability in its findings. This is indicative of a well-structured study design, likely including proper randomization and adherence to intervention protocols, comprehensive data collection without significant missing outcomes, precise outcome measurement, and unbiased reporting of results.

In contrast, the study by Zulman et al., 2019, exhibits a high overall risk of bias, raising concerns about the strength of its conclusions. With high bias due to deviations from intended interventions and concerns in multiple other domains, the internal validity of this study may be compromised, potentially due to changes in the intervention as delivered, issues with data completeness, and the objectivity of outcome measurements. The implications of this study's results should be considered cautiously within the context of its methodological limitations.

Several studies, including those by Grace Sum et al., 2021, Nyweide et al., 2013, Bayliss et al., 2015, Halima Amjad et al., 2016, Aaron Jones et al., 2020, Almaawiy et al., 2014, Nyweide & Bynum, 2017, Menec et al., 2006, Serina et al., 2023, and Lan & Chen, 2022, demonstrate some concerns in the randomization process, which could indicate potential limitations in the random allocation of participants or non-random study designs. However, they generally show low bias in the other domains, suggesting that the conduct of the study and the reporting of results were mostly consistent with the research protocol.

Li Yang et al., 2022, and Bayliss et al., 2015, are noted to have some concerns regarding the bias due to missing outcome data, which could affect the completeness and applicability of the data analysis. However, the low bias in other domains for these studies indicates a reasonable level of confidence in most aspects of their methodology.

The overall risk of bias for the majority of studies falls into the 'some concerns' or 'low' categories, which suggests that while there may be areas of improvement, the evidence provided by these studies contributes valuable insights into the field. However, each study's limitations must be acknowledged when interpreting their findings and applying their findings to practice or further research.

### 3.2. Main outcomes

Based on the studies provided in the extraction table (S1 Dataset) let's explore four themes that emerge from these works. Each theme will be discussed with citations to relevant studies from the table.

**3.2.1 Impact of continuity and coordination of care.**   Studies have demonstrated that continuity and coordination of care, especially for older adults with complex needs or chronic conditions, significantly influence health outcomes. Grace Sum et al. (2021) and Bayliss et al. (2015) both highlight the importance of continuity in care models like the Patient-Centered Medical Home (PCMH) and general continuity of care indices. These studies found associations between continuity of care and improved satisfaction, decreased hospitalization, and better management of chronic conditions [53, 58]. Similarly, Aaron Jones et al. (2020) observed that higher continuity of both primary and specialty physician care was associated with reduced risks of emergency department visits and hospital admissions [60].

**3.2.2 Role of telehealth in managing chronic conditions.**   Telehealth has emerged as a critical component in managing chronic diseases, particularly in remote or underserved populations. Lan & Chen's (2022) study on a telehealth care system for chronic disease management showed that technology acceptance and perceived ease of use significantly influence the adoption and effectiveness of telehealth interventions [66]. Similarly, Serina et al. (2023) discussed the perception of telehealth efficacy among physicians for special populations of older adults, indicating its varying effectiveness based on the type of care needed [65]. These studies underscore the potential of telehealth in enhancing access and continuity of care.

**3.2.3 Quality of life and patient activation.**   Several studies have focused on the outcomes related to the quality of life and patient activation as a result of different care models. Grace Sum et al. (2021) found marginal improvements in needs satisfaction and significant impacts on patient activation measures following the implementation of the PCMH model [53]. Li Yang et al. (2022) reported improvements in the quality of life and various abilities of patients and caregivers in a comprehensive intervention trial for Alzheimer's Disease [57]. These findings suggest that focused interventions can lead to improvements in both the physical and psychological domains of patients and caregivers.

**3.2.4 Health outcomes and system utilization.**   The relationship between care models and health system utilization is a crucial area of study, particularly in understanding how interventions can reduce the burden on healthcare systems. Nyweide et al. (2013) and Almaawiy et al. (2014) both provide evidence that higher continuity of care is associated with a lower rate of preventable hospitalizations and decreased acute care visits in the last weeks of life, respectively [55, 61]. Additionally, Halima Amjad et al. (2016) found that lower continuity of care is associated with higher rates of hospitalization, emergency department visits, and health care spending [59]. These studies collectively highlight the potential for improved care coordination to reduce unnecessary healthcare utilization and costs.

These themes reflect the multifaceted nature of healthcare delivery and its impact on older populations, from the importance of continuity and coordination in care, the emerging role of telehealth, the effects on quality of life and patient activation, to the implications for health outcomes and system utilization. Each study contributes to a broader understanding of how different models and technologies can be effectively employed to improve the care and well-being of older adults.

## 4. Discussion

Continuity and sustainability are critical foundations for delivering high-quality, person-centered care to meet the diverse needs of aging populations. This review aimed to synthesize evidence on the impacts of continuous, sustainable care models in family medicine on outcomes for older adults. The studies demonstrated beneficial associations between continuity of care and reduced acute healthcare utilization, improved chronic disease management, higher satisfaction, and better quality of life. However, fully realizing the benefits of continuous,

sustainable care requires transforming fragmented systems through thoughtful design, system-level support, and a relationship-based philosophy of care.

## 4.1 Continuity of care: A core component of high-quality primary care

Continuity of care with a primary provider and care team is a vital marker of healthcare quality that enables personalized, proactive care aligned with patient goals and values [67]. This review reinforces extensive prior evidence showing greater informational, management, and relational continuity correlates to numerous benefits for patients with complex needs, including older adults. These span reduced hospitalizations and complications of chronic illness improved preventive service delivery, higher satisfaction, and better quality of life [68–71].

Seeing the same provider who knows their priorities and history allows older patients to develop trusting relationships and communicate needs more effectively [72]. Consistent evidence-based monitoring and coordination of services aligned with patient preferences enables better self-management and rapid response to deterioration, avoiding crises [73, 74]. Smooth information transfer across settings prevents "falling through the cracks" during care transitions [75, 76]. At its core, continuity enables providers to see the "whole person" and tailor care accordingly over time [77].

While informational continuity has improved through electronic records, relational and management continuity remain challenged by fragmented systems. Declining primary care time and increasing complexity of care make realizing the benefits of continuity difficult despite clear advantages [78]. Reestablishing continuous healing relationships as the bedrock of care delivery should remain a top policy priority to support healthy aging [79].

Multi-pronged approaches like patient-centered medical homes (PCMHs) show promise in strengthening continuity and coordination [80]. Key elements include cultivating sustained clinician-patient partnerships, care planning, integrated care teams, and enhanced access such as telehealth or expanded hours [63, 81]. But while medical homes emphasize relationship-centeredness, implementation is uneven, and many lack needed resources [82].

Truly prioritizing continuity requires greater investment in primary care infrastructure and workforce [67, 83]. This encompasses care coordination staffing, smaller panel sizes, and reduced administrative burden to allow unhurried visits focused on patients' goals and building relationships [84]. Continuity should be measured and valued as a vital sign. Payment and delivery reforms must move from rewarding discrete services to enabling sustained clinician-patient partnerships [69, 85].

Fostering continuity also requires reducing barriers patients face in consistently accessing care [86]. This includes transportation, flexible scheduling, language access, care navigation, and family caregiver supports–elements that facilitate ongoing engagement. Continuity and community health are interdependent; lasting partnerships connecting clinical and community resources can help address social determinants underlying disease [87].

Ultimately more time, trust, and resources are essential to nurturing continuous healing relationships in primary care [88]. But the value proposition is clear. Continuity provides the foundation for proactive, coordinated, and patient-centered care required for 21st century population health. It should remain central to all efforts to improve care quality, experience, and sustainability for aging populations [89].

## 4.2 Care coordination and team-based models

While primary care continuity provides crucial foundations, optimizing outcomes for complex older adults also requires expanded care coordination and integration [89]. Multidisciplinary

teams who share information and collaborate cross-sectorally can enhance care continuity even when patients see multiple individual physicians [90].

For homebound seniors with intensive needs, programs providing coordinated home-based medical care and social supports have demonstrated reduced hospitalizations, nursing home use, and costs [91]. Models like Geriatric Resources for Assessment and Care of Elders (GRACE) integrate geriatrics expertise into primary care and offer care management by a nurse practitioner and social worker [92].

For high-risk patients, Transitional Care Models coordinate evidence-based care during hospital-to-home transitions via home visits, telehealth, and linking with community services–resulting in fewer rehospitalizations [93]. Embedding social workers in primary care practices helps bridge medical and social needs. And accountable care organizations (ACOs) that share responsibility for entire populations incentivize coordinated, high-value care across settings [94].

These models underscore that while primary care continuity provides the longitudinal relationship, adequately caring for older adults with complex biomedical and social needs requires wider team-based coordination [95]. This review also highlighted the value of care management home visits, cross-setting follow-up, and help navigating systems for high-risk veterans [54].

Still, challenges to team-based care remain, including reimbursement systems that fail to adequately support care management and coordination staff [96]. Colocation and health information exchange facilitate collaboration, but many small practices lack capital and technical assistance for integration [97]. Delivery and payment reforms should accelerate team-based infrastructure development in primary care including provisions for shared complex care management [98, 99].

## 4.3. Health information technology as an enabler

Thoughtfully designed health information technology (HIT) is also essential to enable both informational continuity and care coordination across settings [100]. Universal electronic health records tethered by health information exchange networks can close communication gaps between primary care homes and specialists [101]. Digital interoperability allows up-to-date synthesizable data to follow patients [102].

Virtual technology also increasingly bridges geographical and mobility barriers to continuity. Telehealth expands access and conveniences, as well as enabling remote monitoring of chronic conditions to augment in-person primary care [103]. "Virtual medical homes" provide continuity via digital coordination tools, patient portals, and e-consult platforms linking primary care and specialists [104]. Still, thoughtfulness is required to avoid fragmentation through virtual care [105]. Telehealth should synergize with, not replace, longitudinal human relationships.

HIT offers tools to advance continuity and coordination but is not a panacea. Poor design and implementation stymies progress. Core challenges of relationships and trust require a human focus [106]. Providers face burnout from clerical burdens and usability issues [120]. And patients–especially older adults–may lack digital access or literacy for optimal engagement. HIT integration must carefully balance human needs, clinical workflows, and technical capacities to avoid disruption [107].

While HIT can enable continuity and coordination at scale, it is not a substitute. Progress requires improving technology and its implementation, while simultaneously doubling down on the human foundations of continuous healing relationships and person-centered care planning [108]. Used thoughtfully, HIT provides infrastructure to nurture continuity–but it must serve humanistic healthcare values.

## 4.4 Sustainability and the long view

Realizing the benefits of continuity and coordination requires a long-term sustainability lens. Fragmentation stems partly from systems valuing episodic interventions over lifelong relationships [109]. Innovations like patient-centered medical homes often lack staying power, undermined by waning resources [110].

Sustainable care requires steadfast rather than sporadic commitment to continuity and relationships. This means consistent investment in primary care capacity and workforce [111]. But it also involves broader shifts from short-term utilization metrics toward long-term health outcomes that capture benefits over generations. Value-based payment incentivizing population health helps realign systems [112].

Taking the long view also compels addressing deeper roots of fragmentation like inequities and social determinants. Lasting gains require not just improving healthcare delivery but also strengthening communities [113]. Continuity and community health are interdependent [114].

Environmental sustainability further influences care systems and social determinants [115]. Climate change impacts food security, heat exposure, infectious diseases, disabilities, and mental health [116]. Bioethics highlight sustainability through responsible resource stewardship [117, 118]. An ethics of care prioritizes sustaining relationships and reducing suffering for future generations [119].

To build sustainable care for aging populations, continuity must intertwine with public health and social justice [120].

The path forward is challenging but clear. Continuity, relationships, and care integration remain vital to excellent geriatric care delivery [121]. But sustainable progress requires courageous systems change and social investment [122]. With focus, partnerships, and public engagement, a more relationship-centered, equitable, and resilient future is possible.

## 4.5 Continuity of care and patient safety

Continuity of care stands as a pivotal element in the matrix of patient safety, particularly within the domain of family medicine catering to older populations [123]. It establishes a foundation for a sustained and comprehensive understanding of the patient's health history, preferences, and nuanced needs over time. This continuous thread of familiarity significantly mitigates the risk of medical errors, which are more likely to occur in fragmented systems where patients are seen by multiple, uncoordinated providers [124]. The seamless flow of information that continuity of care ensures can prevent prescription errors, duplicate testing, and conflicting treatments, thereby safeguarding patients from potential harm. Moreover, the ongoing relationship built on trust and mutual understanding enhances patient engagement and compliance, pivotal factors in the effective management of chronic conditions and the prevention of acute episodes that could compromise patient safety [125].

Furthermore, the link between continuity of care and patient safety extends into the broader realm of health outcomes and quality of life for older adults. Through the lens of continuity, healthcare providers can more accurately tailor interventions to the individual's unique health trajectory, optimizing treatment plans and proactively adjusting care to meet evolving needs [126]. This personalized approach not only improves the effectiveness of care but also contributes to a greater sense of security and satisfaction among patients, vital components of quality of life. In essence, continuity of care acts as a safeguard, reducing unnecessary hospitalizations and emergency visits, which are fraught with risks for older adults [127]. By prioritizing and enhancing continuity of care, healthcare systems can thus make significant strides toward a safer, more patient-centered care model that actively contributes to the well-

being and safety of the elderly population, embodying the true essence of comprehensive healthcare [128, 129].

## 5. Conclusion

This systematic review aimed to explore the impacts of continuity and sustainability in family medicine care models on quality of life and health outcomes for older adults. The 14 studies included provide consistent evidence that greater informational, management, and relational continuity of care correlates to reduced healthcare utilization, improved chronic disease management, higher satisfaction, and better quality of life in aging populations. However, fully realizing the benefits of continuous, sustainable care requires addressing systemic fragmentation through thoughtful design, resources, and a relationship-centered philosophy.

Key findings of this review reinforce that continuity should remain a vital sign and central aim of high-quality primary care. Sustained personalized relationships between older patients and primary care providers facilitate proactive, coordinated, and patient-centered care aligned with everyone's priorities. This in turn appears to enable better health maintenance, avoidance of complications requiring hospitalization, and optimization of abilities and quality of life.

Thoughtful system enhancements like patient-centered medical homes, telehealth, care coordination programs, and health information exchange can strengthen continuity and extend its benefits. But technological integration must serve human relationships and meaningfully engage diverse patients. Without attending to relationships and personhood, care risks becoming transactional and fragmented.

At the same time, challenges remain in overcoming ingrained systemic barriers and moving from isolated interventions to lasting transformation. Truly valuing continuity and relationships requires expanded investment in primary care capacity and trusting partnerships. Payment and delivery models must shift focus from discrete utilization metrics toward enabling longitudinal clinician-patient relationships and reducing inequities.

Sustainability involves steadfast rather than sporadic commitment to the foundations of primary care, relationships, and community. It compels aligning health systems with the broader public good including population health, equity, and environmental sustainability. Lasting solutions require courageous systems change and social investment.

This review has limitations including heterogeneity among relatively few studies, self-reported data, and variations in quality and bias risk assessments. The populations and interventions evaluated were also not fully generalizable. Nonetheless, the consistency of observed associations adds to a substantive body of literature on the benefits of continuity.

In conclusion, key implications and recommendations include:

- Continuity of care remains a vital marker of primary care quality and should be universally measured and valued. Policy and payment reforms must incentivize relationship-centered practice.

- Medical homes show potential to strengthen informational, management, and relational continuity but require adequate, sustained financial support and tailored implementation.

- Thoughtfully designed telehealth and HIT tools can augment continuity but should enhance, not replace, human relationships and clinical encounters. These technologies require careful integration to avoid new fragmentation.

- Care coordination models like home-based primary care, GRACE, and transitional care can enhance outcomes for complex older adults when layered upon continuous relationships

with a primary clinician and team. Seamless bi-directional communication between these care team members is essential.

- Truly patient-centered care requires understanding and aligning with each older individual's priorities. Decision-making should be guided by patient values, goals, and preferences.

- High-functioning teams require shared leadership with patients, caregivers, and communities to implement care models focused on whole person health and well-being.

- Expanding the clinical workforce through non-physician team members, telehealth providers, and community health workers builds capacity to support coordinated, continuous care tailored to aging adults' needs.

- Payment and delivery reforms should accelerate team-based infrastructure development in primary care including provisions for shared complex care management and improving social determinants of health.

- A long-term sustainability lens focused on lasting relationships, resilience, and the public good is critical to overcoming fragmentation. Leaders must champion transformative systems change.

Overall, continuity, coordination, and sustainability of patient-centered care models are crucial for improving quality of life and health outcomes for the aging population. While challenges remain, progress toward more humane, equitable, and integrated care is possible through courageous practice transformation and system redesign grounded in humanistic values. This review highlights family medicine's central role in sustaining continuous healing relationships to enable healthy aging. But realizing this vision requires valuing what matters most—the person behind each patient.

## Supporting information

**S1 Dataset. The extraction table.** https://doi.org/10.6084/m9.figshare.25465216.
(DOCX)

**S1 Fig. PRIMISA flow chart.**
(TIF)

**S2 Fig. Risk of bias assessment of included studies.**
(TIF)

## Author Contributions

**Conceptualization:** Mohammed Nasser Albarqi.

**Data curation:** Mohammed Nasser Albarqi.

**Formal analysis:** Mohammed Nasser Albarqi.

**Funding acquisition:** Mohammed Nasser Albarqi.

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
