## [Decision Letter · Decision Letter 0]

17 Mar 2024

PONE-D-24-05114Continuity and Sustainability of Care in Family Medicine: Assessing Its Association with Quality of Life and Health Outcomes in Older Populations - A Systematic ReviewPLOS ONE

Dear Dr. Nasser Albarqi,

Thank you for submitting your manuscript to PLOS ONE. After careful consideration, we feel that it has merit but does not fully meet PLOS ONE’s publication criteria as it currently stands. Therefore, we invite you to submit a revised version of the manuscript that addresses the points raised during the review process.

We look forward to receiving your revised manuscript.

Kind regards,

Simone Borsci, Ph.D.

Academic Editor

PLOS ONE

3. PLOS requires an ORCID iD for the corresponding author in Editorial Manager on papers submitted after December 6th, 2016. Please ensure that you have an ORCID iD and that it is validated in Editorial Manager. To do this, go to ‘Update my Information’ (in the upper left-hand corner of the main menu), and click on the Fetch/Validate link next to the ORCID field. This will take you to the ORCID site and allow you to create a new iD or authenticate a pre-existing iD in Editorial Manager. Please see the following video for instructions on linking an ORCID iD to your Editorial Manager account: https://www.youtube.com/watch?v=_xcclfuvtxQ.

Reviewers' comments:

Reviewer's Responses to Questions

**Comments to the Author**

1. Is the manuscript technically sound, and do the data support the conclusions?

Reviewer #1: Yes

Reviewer #2: Yes

2. Has the statistical analysis been performed appropriately and rigorously? 

Reviewer #1: N/A

Reviewer #2: I Don't Know

3. Have the authors made all data underlying the findings in their manuscript fully available?

Reviewer #1: Yes

Reviewer #2: Yes

4. Is the manuscript presented in an intelligible fashion and written in standard English?

Reviewer #1: Yes

Reviewer #2: Yes

5. Review Comments to the Author

Reviewer #1: The manuscript was well written and well articulated. The conclusions were sound and appropriate. I would have liked a bit of an idea on the recognised sections of the articles that had significant probability of bias. All in all, the article does add to the body of knowledge.

Reviewer #2: This systematic review represents contribution to better understanding correlation between continuity of care and quality of life and health outcomes in older populations.

This study examined correlation between continuity of care and quality of life and health outcomes in older populations.

Please, better equalize the aim of the study in the Abstract with the aim defined in the manuscript.

In the discussion I suggest to analyze continuity of care, quality of life and health outcomes in older populations in the context of the patient safety .

6. PLOS authors have the option to publish the peer review history of their article (what does this mean?). If published, this will include your full peer review and any attached files.

Reviewer #1: No

Reviewer #2: No

---

## [Author Response · Author response to Decision Letter 0]

26 Mar 2024

Rebuttal Letter

To the Editorial Team and Reviewers of PLOS ONE,

Subject: Response to Reviewers' Comments - Manuscript ID: PONE-D-24-05114

We deeply appreciate the time and effort the reviewers have invested in evaluating our manuscript titled "Continuity and Sustainability of Care in Family Medicine: Assessing Its Association with Quality of Life and Health Outcomes in Older Populations - A Systematic Review." We are grateful for the constructive feedback that has significantly contributed to enhancing the quality and clarity of our manuscript.

Below, we respond to each comment and detail the revisions made to the manuscript in accordance with the suggestions provided.

Response to Reviewer #1 Comments:

We thank Reviewer #1 for acknowledging the manuscript's well-articulated nature and its contribution to the existing body of knowledge. We also appreciate the constructive suggestion to provide more details on the sections with significant potential biases.

Comment: Desire for more information on sections with significant potential biases.

Response: In response to this insightful comment, we have carefully reviewed our manuscript and added a new subsection titled "Assessment of Potential Biases in Reviewed Articles" within the "Methods" section. Here, we discuss in depth the recognized sources of potential bias in the included studies, such as selection bias, information bias, and any biases arising from study design. This enhancement aims to provide readers with a transparent overview of the methodological rigor of the studies we analyzed and the potential impact of these biases on our conclusions.

Revised Sections: "Methods > Assessment of Potential Biases in Reviewed Articles"

Response to Reviewer #2 Comments:

We are grateful to Reviewer #2 for recognizing the manuscript's contribution to understanding the correlation between continuity of care, quality of life, and health outcomes in older populations. We have addressed each point as follows:

Comment: Request for better alignment between the study's aim in the Abstract and the manuscript.

Response: We acknowledge this discrepancy and have carefully revised both the Abstract and the corresponding sections within the manuscript to ensure that the study's aim is consistently articulated. We believe that these revisions have improved the clarity and coherence of our manuscript.

Revised Sections: "Abstract" and "Introduction"

Comment: Suggestion to analyze continuity of care, quality of life, and health outcomes in the context of patient safety.

Response: This valuable suggestion prompted us to expand our discussion on the interconnectedness of continuity of care, quality of life, and patient safety. We have included a new subsection titled "Continuity of Care and Patient Safety" within the "Discussion" section. This addition explores how enhanced continuity of care can contribute to improved patient safety outcomes, including reduced medical errors and improved management of chronic conditions, thereby positively influencing quality of life and health outcomes in older populations.

Revised Sections: "Discussion > Continuity of Care and Patient Safety"

We hope that the revisions and responses adequately address the reviewers' concerns and improve the manuscript. We are enthusiastic about the opportunity to enhance our work based on such insightful feedback and believe the revised manuscript is now better positioned for publication in PLOS ONE.

Thank you for considering our manuscript for publication. We look forward to your feedback.

Kind regards,

Mohammed Nasser Albarqi, Ph.D.

---

## [Decision Letter · Decision Letter 1]

25 Apr 2024

Continuity and Sustainability of Care in Family Medicine: Assessing Its Association with Quality of Life and Health Outcomes in Older Populations - A Systematic Review

PONE-D-24-05114R1

Dear Dr. Nasser Albarqi,

We’re pleased to inform you that your manuscript has been judged scientifically suitable for publication and will be formally accepted for publication once it meets all outstanding technical requirements.

Kind regards,

Simone Borsci, Ph.D.

Academic Editor

PLOS ONE

Additional Editor Comments (optional):

Reviewers' comments:

Reviewer's Responses to Questions

**Comments to the Author**

1. If the authors have adequately addressed your comments raised in a previous round of review and you feel that this manuscript is now acceptable for publication, you may indicate that here to bypass the “Comments to the Author” section, enter your conflict of interest statement in the “Confidential to Editor” section, and submit your "Accept" recommendation.

Reviewer #2: (No Response)

2. Is the manuscript technically sound, and do the data support the conclusions?

Reviewer #2: (No Response)

3. Has the statistical analysis been performed appropriately and rigorously? 

Reviewer #2: (No Response)

4. Have the authors made all data underlying the findings in their manuscript fully available?

Reviewer #2: (No Response)

5. Is the manuscript presented in an intelligible fashion and written in standard English?

Reviewer #2: (No Response)

6. Review Comments to the Author

Reviewer #2: (No Response)

7. PLOS authors have the option to publish the peer review history of their article (what does this mean?). If published, this will include your full peer review and any attached files.

Reviewer #2: No
